# Injectable Nanomedicine–Hydrogel for NIR Light Photothermal–Chemo Combination Therapy of Tumor

**DOI:** 10.3390/polym14245547

**Published:** 2022-12-18

**Authors:** Dashan Qi, Haowei Zhu, Yingjie Kong, Qingming Shen

**Affiliations:** State Key Laboratory of Organic Electronics and Information Displays & Jiangsu Key Laboratory for Biosensors, Institute of Advanced Materials (IAM), Nanjing University of Posts & Telecommunications, Nanjing 210023, China

**Keywords:** injectable hydrogel, nanomedicine, near-infrared, photothermal therapy, chemotherapy

## Abstract

Traditional hydrogels have drawbacks such as surgical implantation, large wound surfaces, and uncontrollable drug release during tumor treatment. In this paper, targeted nanomedicine has been combined with injectable hydrogel for photothermal–chemotherapy combination therapy. First, targeted nanomedicine (ICG—MTX) was fabricated by combining near-infrared (NIR) photothermal reagents (ICG) and chemotherapy drugs (MTX). The ICG—MTX was then mixed with the hydrogel precursor and radical initiator to obtain an injectable hydrogel precursor solution. Under the irradiation of NIR light, the precursor solution could release alkyl radicals, which promote the transition of the precursor solution from a liquid to a colloidal state. As a result, the nanomedicine could effectively remain at the site of the tumor and continue to be released from the hydrogel. Due to the targeted nature of MTX, the released ICG—MTX could target tumor cells and improve the accuracy of photothermal–chemo combination therapy. The results indicated that the injectable nanomedicine–hydrogel system has a favorable therapeutic effect on tumors.

## 1. Introduction

Cancer is a disease caused by abnormal cells and can spread to different tissues and organs [1,2]. The prevention and treatment of cancer have become an important research topic in the medical profession [3]. Chemotherapy has been widely used to treat cancer [4,5,6]. However, the systemic application of chemotherapy drugs is often accompanied by serious side effects, including obvious toxicity and drug resistance to normal organs, which limits the improvement of clinical symptoms [7]. Therefore, the development of a preferable strategy for cancer treatment is highly expected.

Conventional chemotherapy usually involves high doses and repeated administration to obtain the necessary therapeutic effects, which leads to poor patient compliance, low overall efficacy of the drug, and serious side effects [8,9,10]. In order to overcome these problems, researchers have introduced drug release systems to control the availability of drugs to cancer cells and tissues for a long period of time [11]. At present, nanocarriers, such as nanoparticles, liposomes, etc., have been wildly investigated for chemotherapy and even combination therapies [12,13]. However, due to the disadvantages of nanoparticles, such as low therapeutic accuracy, short residence time, and poor biocompatibility, it is necessary to design a better drug carrier to solve these problems [14,15,16]. As one of the promising drug carriers, hydrogels have been widely studied because of their long-term accumulation and release of drugs at the tumor site, low toxicity of systemic drugs, and controlled release of drugs. Therefore, they are considered one of the best drug delivery systems for tumor chemotherapy [17].

Among the hydrogels, injectable hydrogels have aroused great interest because of their significant physical properties, controllable degradability, continuous drug release, and minimally invasive drug administration process [18]. Especially photoresponsive hydrogels, as emerging biological materials, have been wildly used in photothermal/photodynamic therapy, regenerative medicine, and so on [19,20]. Ultraviolet and visible light can trigger photochemistry to remotely control the physical and chemical reactions of hydrogels. However, because of the shallow penetration depth of ultraviolet and visible light and the killing effect of UV on normal tissues, they have limited application in deep tissue therapy and in situ regenerative medicine [21]. In contrast, deeper penetration depth of near-infrared (NIR) light can effectively penetrate thicker biological samples without affecting surrounding cells and can be widely used in the field of tumor treatment. Therefore, the antitumor impact of the light-responsive hydrogel can be improved by using NIR light [22,23]. 

In recent years, the use of photothermal therapy (PTT) in conjunction with chemotherapy has emerged as a possible therapeutic approach [24]. Not only does it retain the advantages of PTT with strong specificity, low systemic toxicity, and strong controllability, but it overcomes the disadvantages of the poor effect of chemotherapy treatment. Herein, we prepared a hydrogel drug-carrying system with a tumor-targeting effect, as shown in Figure 1. Methotrexate (MTX), a chemotherapy medication, and folic acid have remarkably similar chemical structures. Since folic acid receptors are overexpressed on the surface of cancer cells, Methotrexate is a chemotherapeutic medication to prevent deoxyribonucleic acid synthesis as well as a “targeted ligand” to bind to certain receptors [25]. Inspired by this binding affinity, ICG—MTX was fabricated by combining MTX with NIR photothermal reagent indocyanine green (ICG) through assembly, which can combine the tumor self-recognition capabilities of nanomedicines with the synergistic photothermal chemotherapy effects. Carrier-free nanomedicines are a promising cancer treatment solution [26]. Nevertheless, nanomedicine still has issues with tumor cell absorption efficiency and the body’s limited blood circulation [27]. In order to overcome the above disadvantages, by mixing ICG—MTX with monomer and initiator in an aqueous solution, a hydrogel precursor solution was prepared. The integration of ICG—MTX nanomedicines into hydrogel precursor could introduce the NIR photothermal function of ICG—MTX. In the liquid state, it can be injected in situ through a syringe because of its superior fluidity. The photothermal performance of ICG—MTX could stimulate the decomposition of AIPH (2,2-azobis [2-(2-imidazolin-2-yl) propane] dihydrochloride, a heat-sensitive molecule, can be decomposed under heating conditions to generate alkyl radicals, which can destroy a variety of cellular components under anaerobic conditions) and release free radicals, and can trigger the transformation from solution to gel state (sol–gel), and form the ICG—MTX@PPA (PNIPA, PEGDA, and AIPH) Gels [28]. Because of the porous three-dimensional network structure of the hydrogel, the nanomedicine (ICG—MTX) could be released from ICG—MTX@PPA Gels by permeation effect and the nanomedicine could diffuse slowly to the tumor cells with folic acid receptors (Hela, 4T1, etc.) for photothermal therapy and chemotherapy [29]. This multifunctional nanomedicine-hydrogel system has great biocompatibility, NIR photothermal therapy, ICG—MTX targeting, and intelligent response. By enhancing the accumulation of tumor medicines and enhancing the precision of treatment, the system is anticipated to increase the antitumor efficacy.

## 2. Methods

### 2.1. Materials

All compounds have reached the analytical level and do not need to be further purified. Polyethylene glycol diacrylate (PEGDA), methylene blue (MB), and acrylamide (AM) were purchased from Macklin. Methotrexate (MTX), sodium hydroxide, indocyanine green, N,N,N′,N′-tetramethylethylenediamine, and 3-(4,5-dimethyl-2-thiazolyl)-2,5-diphenyl tetrazolium bromide were acquired from Aladdin. N-isopropylacrylamide was acquired from Acme.

### 2.2. Characterization

The morphology of nanoparticles was observed using a transmission electron microscope (TEM, HT7700, Hitachi, Tokyo, Japan) with an acceleration voltage of 100 KV. A scanning electron microscope (SEM, SU3800, Hitachi, Tokyo, Japan) was used to observe the morphology of nanoparticles. The size of nanoparticles was measured using a laser particle size meter (Zeta PALS, Brookhaven, New York, NY, USA). A Shimadzu UV-3600 spectrophotometer was utilized to record the absorption spectra of our samples at room temperature (Lambda 650, PerkinElmer, Waltham, MA, USA). A fluorescence spectrometer was used to measure the fluorescence of nanoparticles (RF6000, shimadzu, Kyoto, Japan). The 808 nm laser was purchased from Changchun New Industries Optoelectronics Technology Co., Ltd. Using a rheometer with parallel plates, the dynamic rheology of nanoparticles was studied (Discovery HR 20, TA Instruments, New Castle, DE, USA). An infrared camera (FLIR E50, FLIR Systems Inc., Wilsonville, OR, USA) was used to record the temperature changes during the illumination of the sample. A PowerWave XS/XS2 microplate spectrophotometer (SynergyHTX, BioTek, Winooski, VT, USA) was used for the methyl thiazolyl tetrazolium (MTT) analysis. Flow cytometry (Flow Sight, 2040R, Merck Millipore, Burlington, MA, USA) was used to observe the cell transfer under different conditions. The survival of stained cells was observed by fluorescence microscopy (CLSM, FV1000MPE, Olympus, Kyoto, Japan).

### 2.3. Preparation and Characterization of ICG—MTX

After mixing 1.0 mL of ICG solution (1.0 mg·mL^−1^) with 1.0 mL of MTX solution (1.0 mg·mL^−1^), the pH value of the solution was adjusted to 7 with 1.0 mol·L^−1^ NaOH. After aging for 24 h, the solution was ultrafiltration at a speed of 3000 rpm for 15 min (with a molecular weight of 10 kDa) [30].

The morphology of ICG—MTX was measured by TEM. UV/visible/near-infrared spectrophotometers and fluorescence, respectively, were used to measure UV–visible (UV–vis) absorption and near-infrared fluorescence intensity.

### 2.4. Photothermal Performance Test of ICG—MTX

The temperature of different concentrations of ICG and ICG—MTX (ICG content is 0, 16.5, 31.3, 62.5, and 125 µg·mL^−1^) were measured for 5 min under laser irradiation (808 nm, 0.5 W·cm^−2^), and then an infrared camera was used to record the shift in temperature. Additionally, the photothermal effects of ICG—MTX (31.3 µg·mL^−1^) were investigated at various power intensities (0.3, 0.5, 0.7, and 1.0 W·cm^−2^).

An 808 laser with 0.5W/cm^2^ was used to irradiate the ICG solution (200 mL, 31.3 μg·mL^−1^) until it reached a stable state. Equation (1) is used to calculate the photothermal conversion efficiency (η):(1)$$\eta =\frac{hS\left(T_{max}-T_{surr}\right)-Q_{Dis}}{I\left(1-10^{A_{\lambda }}\right)}\;$$
where *h* is the heat transfer coefficient, *S* is the surface area of the container, *T_Surr_* is the ambient temperature, *T_Max_* is the highest temperature at which the solution reaches stability under irradiation of 808 nm laser, *I* is the power density of the 808 nm laser (0.5 W·cm^−2^), *A*_λ_ is the absorbance of ICG—MTX at 808 nm, and *Q_Dis_* is the heat loss of the light absorbed by the quartz sample cell. Equation (2) is used to calculate *hS*:(2)$$hS=\frac{m_{D}c_{D}}{\tau_{s}}\;$$
where *m_D_* is the mass of solvent-deionized water and *c_D_* is the heat capacity of deionized water (4.2 J·g^−1^). Meanwhile, Equation (3) is used to calculate the sample system time constant of *τ_s_*:(3)$$t=-\tau_{s}ln\theta$$
where *t* is the time point corresponding to different temperatures of the solution after cooling from the highest temperature. Equation (4) is used to calculate *θ*:(4)$$\theta =\frac{T-T_{Surr}}{T_{Max}-T_{Surr}}$$
where *T* is the temperature corresponding to each time point during the cooling process.

### 2.5. Detection of ABTS+·Free Radicals

The mixed solutions of ABTS (2.0 mg·mL^−1^, 0.2 mL) and AIPH (0.2 mg·mL^−1^, 0.2 mL) were sampled at different time points at 37 or 45 °C to be tested (2, 4, and 6 h) under dark conditions.

The production of ABTS+ is performed by using the interaction of ABTS and AIPH to produce toxic free radicals. The mixed solution of ABTS (2.0 mg·mL^−1^, 0.2 mL) and DGA NPs (1.0 mg·mL^−1^, 0.2 mL) was sampled for testing after 10 min with or without 808 nm laser irradiation. Subsequently, it was analyzed using the ultraviolet–visible spectrum.

### 2.6. Preparation of Hydrogel Monomer PNIPAM

PNIPAM was prepared according to the reported method. N-isopropyl acrylamide (NIPAM), acrylamide (AM), and N,N′-bisacryloylcysteamine (BAC) have a molar ratio of 9: 1: 0.2, respectively. For instance, prepare an aqueous solution containing 1.017 g NIPAM and 0.0071 g AM by first dissolving 0.052 g BAC in 70% ethanol. In order to eliminate any dissolved oxygen, add nitrogen to the mixture after adding 300 µL of N,N,N′,N′-tetramethylethylenediamine (TEMED) and let it bubble for 30 min. Then add 600 µL of freshly made 10% *w*/*v* ammonium persulfate to start the polymerization process, which is then left to run overnight at room temperature. Remove the unreacted monomer for two days while the hydrogel is placed in 0.5 L of deionized water for dialysis. Then freeze-dry and set aside.

### 2.7. Preparation and Characterization of ICG—MTX@PPA Gels

After mixing 0.339 g PNIPAM, 2.0 mL PEGDA, 1.0 mL AIPH (10% *w*/*v*), and ICG—MTX solution evenly, heat it at a temperature above 45 °C or under 808 nm light to form a hydrogel. Use the test tube inversion approach to detect the composite material ICG—MTX@PPA Gels’ sol–gel phase transition.

SEM was used to examine the morphology of ICG—MTX@PPA Gels. In a nutshell, the hydrogel is broken down and frozen in liquid nitrogen, and gold is then applied to the sample slice’s surface in preparation for SEM imaging.

A parallel-plate rheometer (Discovery HR 20) was used to measure the dynamic rheology of ICG—MTX@PPA Gels. The diameter of ICG—MTX@PPA Gels (15 wt.%) is 4 cm, and the gap is 0.5 mm. Changes in the hydrogel’s viscosity, storage modulus (G′), and loss modulus (G″) were observed when the heating rate was 0.5 °C/min, and the data were gathered at 1.0 Hz.

### 2.8. Release Behavior of ICG—MTX Nano Particles from ICG—MTX@PPA Gels

In order to study the drug release property of the hydrogel, the prepared ICG—MTX@PPA Gels were immersed into 5.0 mL phosphate buffer saline (pH = 6.5) and incubated in a shaker at 37 °C or 50 °C (100 rpm). After a certain amount of time, 400 µL of release medium was replaced with the same volume of fresh PBS.

The release of ICG—MTX was detected by a fluorescence spectrophotometer. The cumulative release percentage is calculated as follows:(5)$$E_{r}\left(\%\right)=\frac{F_{t}}{F_{100}}\times 100\%$$

### 2.9. Cell Experiment

We selected 3T3 and Hela as cell models and cultured them at 37 °C and 5% CO_2_ in a DMEM medium containing 10% FBS. The cells were cultured at 37 °C for 24 h with a complete medium in a 96-well plate (1 × 104 cells per well). The cells were then treated with different concentrations of ICG—MTX and ICG—MTX@PPA Gels in a fresh medium. After being cultured for 12 h, MTT (20 µL) was added to each well and further cultured for 4 h. Next, after removing the supernatant, the methyl group formed was dissolved with dimethyl sulfoxide, and the untreated cells were used as a control, and their absorbance was monitored by a multifunctional enzyme marker.

Transwell-24 co-culture technique (Corning) was used to assess the antitumor efficacy of ICG—MTX@PPA Gels in vitro. The 24-well Transwell chamber received 500 µL of Hela cell suspension, which was then grown there for 24 h. The upper inserts were added to a 24-well plate of cells and co-cultured for 12, 24, and 48 h, and then the MTT test was performed. After that, the Hela cells were divided into five groups and treated with different materials, I. PBS, II. ICG—MTX, III. ICG—MTX with laser, IV. ICG—MTX@PPA Gels, V. ICG—MTX@PPA Gels with laser. The 808 nm laser was used in the laser-treatment group for 5 min at a power density of 0.5 W/cm^2^. Cells were cultured for 6, 12, and 24 h before the MTT test was carried out. The FITC/PI apoptosis kit was used to perform a flow cytometric analysis of the cytotoxicity.

In order to distinguish living and dead cells, each group of cells was stained with calcein (AM, 4 µM) and propidine iodide (PI, 4 µM) for 30 min. An inverted fluorescence microscope was then used to observe its fluorescence.

## 3. Results and Discussion

### 3.1. Preparation and Characterization of ICG—MTX

ICG—MTX was constructed by self-assembly of the targeted chemotherapy drug MTX and the photothermal reagent ICG. ICG and MTX were combined into ICG—MTX nanoparticles through a combination of electrostatic contact, hydrophobic action, and stacking. The morphology and size of ICG—MTX were observed by transmission electron microscopy (TEM) and dynamic light scattering (DLS); from Figure 2a, the shape of ICG—MTX is spherical, and its hydrodynamic particle size is about 160 nm. The absorption intensity of ICG and ICG—MTX in an aqueous solution increases linearly with the concentration increase (Figure 2a–d). As shown in Figure 2b, after ICG and MTX are self-assembled together, the absorption peak at 780 nm shifted to 790 nm, indicating that there is a π–π accumulation interaction between the aromatic ring of ICG and the aromatic ring of MTX. The results show that various weak interactions, such as static electricity and hydrophobicity/π–π stacking, are the main driving forces for the self-assembly of the nanomedicine. As shown in Figure 2c, ICG and ICG—MTX exhibit similar fluorescence emission peaks, indicating that ICG has been successfully assembled with MTX. As is well known, ICG has a chemically and conformationally unstable olefin structure [27]. The enhanced fluorescence intensity of nanomedicines can be attributed to the spatial steric resistance of the adjacent molecule MTX, alleviating the irregular aggregation of ICG. For Figure 2d, ICG—MTX is slowly released from the hydrogel over time, and the release rate at 50 °C is slightly higher than 37 °C. The result shows that the structure of hydrogel is a porous three-dimensional, which can be used as a sustained-release drug carrier at a high treatment temperature (50 °C) to slowly release the drug.

### 3.2. NIR-II Photothermal Performance of ICG—MTX

The infrared thermal cameras were used to record the temperature shift of ICG and ICG—MTX under irradiation. From Figure 3a–d, ICG—MTX shows excellent photothermal conversion ability and near infrared thermal imaging function. When ICG (31.3 µg·mL^−1^) and ICG—MTX (ICG, 31.3 µg·mL^−1^) were irradiated under 808 nm laser (0.5 W·cm^−2^) for three ON/OFF cycles, the photothermal performance of ICG decreased sharply, while the photothermal performance of ICG—MTX solution was almost unchanged. That means that ICG—MTX has better photothermal stability (Figure 4a,b) than ICG alone. In addition, according to the evaluation of photothermal data, the photothermal conversion efficiency (η) of ICG—MTX is 34.9% (Figure 4c), which is 30.7% higher than the photothermal conversion efficiency (26.7%) of ICG (Figure 4d). The above results indicate that ICG—MTX has an excellent light absorption capacity, good biocompatibility, superior NIR photothermal conversion capability, and great photothermal stability. ICG—MTX nanomedicine showed improved photothermal stability and photothermal conversion efficiency, which means that it will have good application potential in PTT.

### 3.3. Detection of ABTS+·Free Radicals

2,2′-Azinobis-(3-ethylbenzthiazoline-6-sulphonate) (ABTS) was used as a chemical probe because the non-characteristic absorption of ABTS can quickly react with the free radicals produced from AIPH to form 2,2′-diazo (3-ethylbenzothiazoline-6-sulfonic acid (ABTS+). The resulting ABTS+· exhibits a typical UV–vis absorption spectrum in the range of 500–900 nm. The decomposition ability of AIPH was studied by evaluating the concentration of ABTS+. When ABTS was treated with ICG—MTX at different temperatures, the production of ABTS+· related to the incubation time was investigated. As shown in Figure 5a, the concentration of ABTS+ at 45 °C is significantly higher than that of 37 °C, which indicates that AIPH will decompose rapidly when the temperature rises. As shown in Figure 5b, AIPH could decompose and produce more alkyl radicals (R·) when the ICG—MTX nanomedicine was irradiated under an 808 nm laser.

### 3.4. Preparation and Characterization of ICG—MTX@PPA Gels

Hydrogels formed in situ are beneficial to the treatment of cancer and have been extensively studied [31]. In order to develop injectable hydrogels, PNIPAM, PEGDA, and AIPH were introduced. First, PNIPAM was synthesized according to the previously reported preparation method [32]. After mixing PNIPAM, PEGDA, and AIPH evenly, a precursor solution of PPA Gels was obtained. When the temperature of PPA Gels reaches above 45 °C, the alkyl radicals R release from AIPH could induce PEGDA and PEGDA to cross-link with each other. As shown in SEM images (Figure 6b,c), the hydrogel showed a three-dimensional porous structure, and the pore size ranges from 1 μm to 4 μm.

The ICG—MTX nanomedicine was then mixed with PNIPAM, PEGDA, and AIPH, to obtain a precursor solution of ICG—MTX@PPA Gels. Under laser irradiation, the high temperature generated by ICG—MTX could induce the decomposition of AIPH to produce R· and then trigger the gelation of PPA Gels. The images in Figure 6e, f confirm the network structure of the ICG—MTX@PPA Gels hydrogel, which has a higher uniform pore size than the PPA Gels, and the pore size is about 2.5 μm. This indicates that the addition of ICG—MTX nanomedicine can enhance the bonding between the hydrogels.

In addition, dynamic rheology experiments have been performed on PPA Gels and ICG—MTX@PPA Gels to quantitatively study the changes in mechanical properties during gelation. At room temperature, the storage modulus and loss modulus of these two hydrogels is very low (<1 Pa), which shows that the fluidity and injectability of the hydrogels are extremely excellent. However, the storage modulus and loss modulus of PPA Gels suddenly increased by more than three orders of magnitude after 6 min when the temperature was set to 45 °C, indicating the formation of hydrogel (Figure 7a). As shown in Figure 7b, there is a similar change in the viscosity, which also indicates the formation of a gel state. Compared with PPA hydrogel, the glue-forming speed of ICG—MTX@PPA hydrogel is slightly reduced, and the G′ decreases in the first 800 s (Figure 7c); similar results can be obtained from changes in the viscosity of the hydrogel. The viscosity of ICG—MTX@PPA hydrogel increases with time, and the growth rate is slower than that of PPA hydrogel; however, the gel could still be formed properly in about 10 min (Figure 7d).

### 3.5. Controlled Drug Release of ICG—MTX@PPA Gels

The release experiment of ICG—MTX from ICG—MTX@PPA Gels in vitro was performed in a buffer solution with pH = 6.8 (PBS), which was used to simulate the tumor microenvironment. From Figure 2d, ICG—MTX nanomedicines slowly and continuously released from the hydrogel. The release speed of DGA NPs was faster within 24 h, and the release speed became slower in the next few days. After 78 h of culture, the cumulative release is about 76%. These results preliminarily indicate that the ICG—MTX@PPA Gels hydrogel system is injectable and biodegradable, with sustained drug release, which is necessary for long-term topical administration.

### 3.6. Cell Experiment

Good NIR photothermal effect and photoinitiated hydrogel drug release properties can be used in PTT and chemotherapy combined therapies. The 3T3 and Hela were selected as cell models to determine the antitumor properties of the ICG—MTX@PPA Gels. The cell viability was first quantitatively studied using the standard 3-(4,5-Dimethyl-2-thiazolyl)-2,5-diphenyl-2H-tetrazolium bromide (MTT) assay. The results show that ICG—MTX and ICG—MTX@PPA Gels are not cytotoxic to normal cells 3T3 but are slightly toxic to Hela cells. With the increased concentration of ICG—MTX or ICG—MTX@PPA Gels, the cell survival rate is also declining, indicating ICG—MTX nanomedicines have slight toxicity to tumor cells (Figure 8a).

The treatment groups were divided into the following five groups: I. PBS, II. ICG—MTX, III. ICG—MTX@PPA Gels, IV. ICG—MTX with laser and V. ICG—MTX@PPA Gels with laser. As can be seen from Figure 8b, the dark toxicity of ICG—MTX@PPA Gels to Hela is inferior to that of ICG—MTX. Since the encapsulated ICG—MTX will be slowly released from the hydrogel over time, ICG—MTX@PPA gel shows a higher cell survival rate than ICG—MTX, and the cell survival rate can reach 95%. In the 808nm laser irradiation group, the cell survival rate after treating Hela with ICG—MTX and ICG—MTX@PPA for 48 h were about 40% and 10%, respectively. The cell survival rate of the ICG—MTX@PPA Gels group was the lowest, which also shows that ICG—MTX@PPA Gels has superior antitumor activity and is a promising nanoplatform for treatment.

After membrane-linked protein V-FITC/PI staining, the flow cytometry was used to study the apoptosis effects of ICG—MTX and ICG—MTX@PPA Gels on Hela cells. Neither ICG—MTX nor ICG—MTX@PPA Gels showed significant cytotoxicity without the irradiation of the NIR laser.

However, groups treated with either ICG—MTX or ICG—MTX@PPA Gels under laser irradiation had significantly higher apoptosis rates, with the ICG—MTX@PPA Gels group having the highest apoptosis rate of 93.66%. This indicates that ICG—MTX@PPA Gels have a good antitumor effect (Figure 9).

In order to evaluate the efficacy of photothermal therapy in vitro, an inverted fluorescence microscope was used to visually distinguish cell vitality. Before observation, all cells were incubated with the above materials for 24 h and then irradiated with an 808 nm laser (0.5 W·cm^−2^) for 10 min, and the cells were stained with calcein-AM and propidine iodide (PI) dyes. The green fluorescence of calcein-AM indicates living cells, and the red fluorescence of PI indicates dead cells. In the ICG—MTX@PPA Gels with the laser treatment group, cell necrosis or apoptosis is the highest (Figure 10), indicating that ICG—MTX@PPA Gels have efficient antitumor properties.

## 4. Conclusions

In conclusion, an injectable ICG—MTX@PPA Gel has been developed by self-assembly. ICG—MTX@PPA Gels can exhibit a sol–gel transition under photothermal stimulation, resulting in the accumulation at the tumor site and controlled release of nanomedicine. Moreover, the multifunctional nanomedicine–hydrogel system shows additional excellent properties, such as NIR photothermal therapy and controlled release of the chemotherapy drug (MTX). The ICG—MTX@PPA Gels show promising antitumor efficacy in combination with phytotherapy and chemotherapy therapy. This work provides a promising strategy to achieve synergistic therapy of tumors.

## Figures and Tables

**Figure 1 polymers-14-05547-f001:**
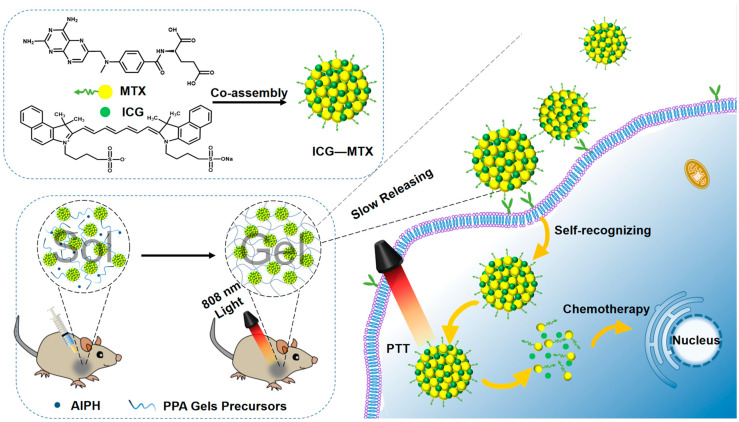
The preparation of ICG—MTX and ICG—MTX@PPA Gels and their application in tumors therapy.

**Figure 2 polymers-14-05547-f002:**
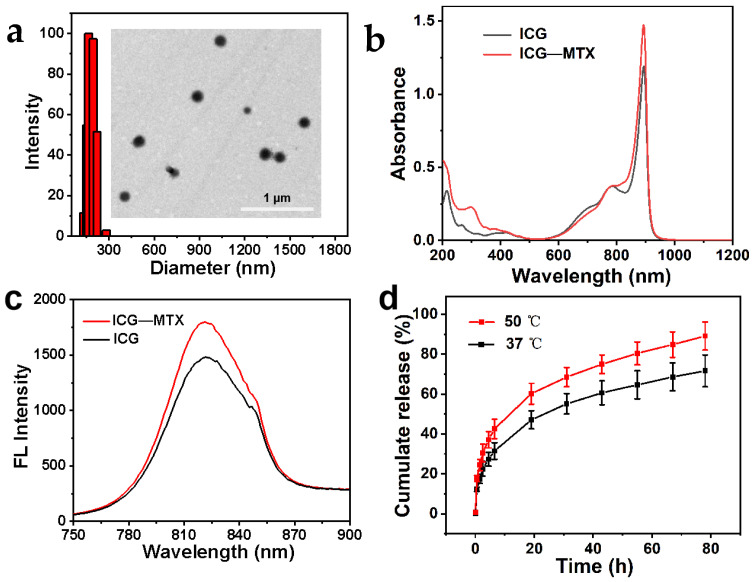
(**a**) ICG—MTX TEM pictures and DLS data. (**b**) ICG—MTX and ICG’s ultraviolet, visible, and infrared absorption spectra. (**c**) Fluorescence spectra of ICG—MTX and ICG. (**d**) ICG—MTX is released from CG–MTX@PPA Gels.

**Figure 3 polymers-14-05547-f003:**
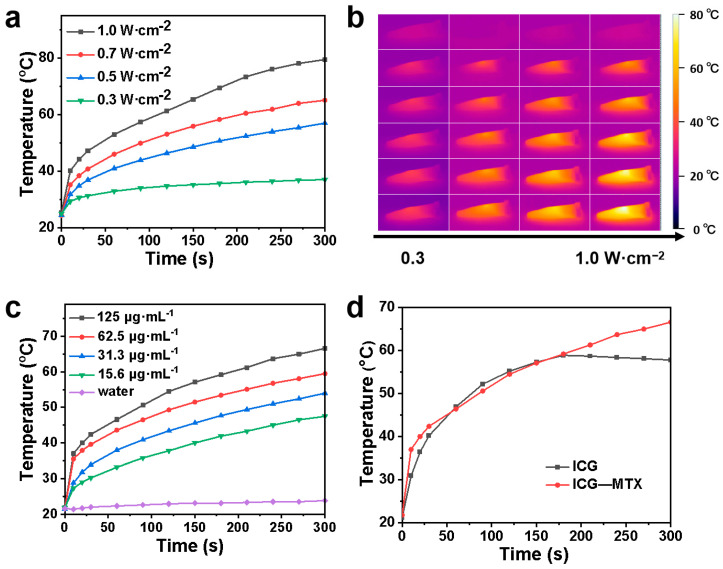
The photothermal performance of ICG—MTX. (**a**) The photothermal performance of ICG—MTX (31.3 µg·mL^−1^) under irradiation of 808 nm lasers with different laser power intensities. (**b**) Infrared thermal images of ICG—MTX (ICG, 31.3 µg·mL^−1^) under irradiation of different laser power of 808 nm laser. (**c**) The photothermal performance of different concentrations of ICG—MTX under 808 nm laser irradiation at 0.5 W·cm^−2^. (**d**) The photothermal effect of ICG (31.3 µg·mL^−1^) and ICG—MTX (ICG, 31.3 µg·mL^−1^) solutions under 808 nm laser irradiation (0.5 W·cm^−2^).

**Figure 4 polymers-14-05547-f004:**
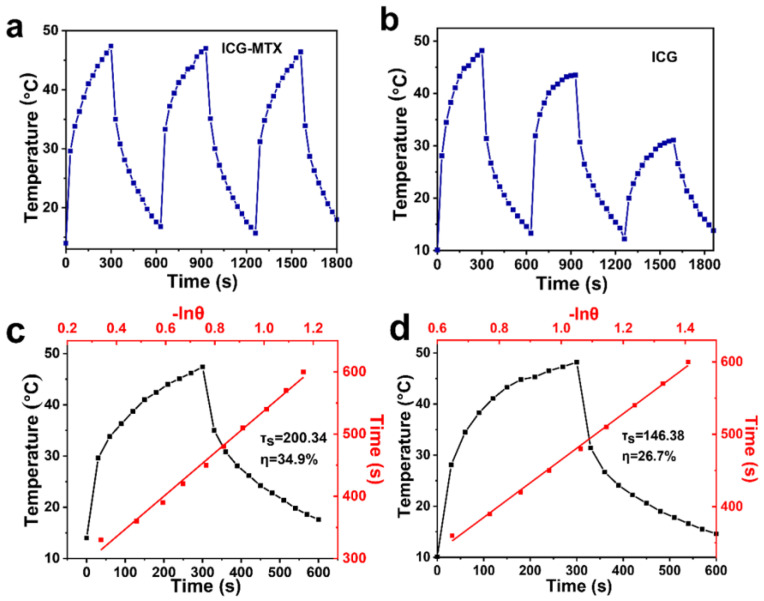
The photothermal stability of ICG—MTX (**a**) and ICG (**b**), and the photothermal conversion efficiency of ICG—MTX (**c**) and ICG (**d**).

**Figure 5 polymers-14-05547-f005:**
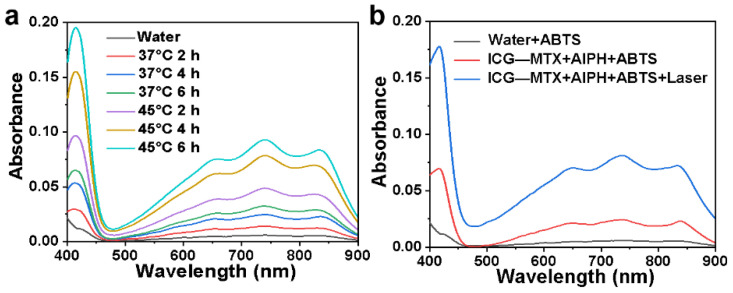
(**a**) The production of ABTS+· induced by free radicals released from AIPH at different temperatures. (**b**) the production of ABTS+· caused by free radicals released from AIPH and ICG—MTX, with/without laser irradiation.

**Figure 6 polymers-14-05547-f006:**
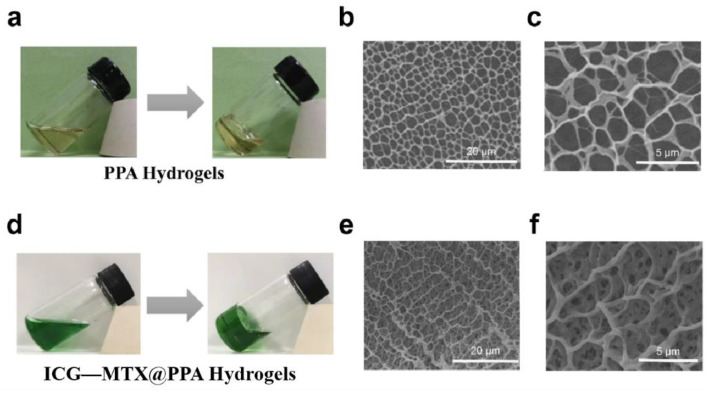
(**a**) The sol–gel transition of PPA Gels at 45 °C and its SEM image (**b**,**c**). (**d**) The sol–gel transition of ICG—MTX@PPA Gels under laser irradiation and its SEM image (**e**,**f**).

**Figure 7 polymers-14-05547-f007:**
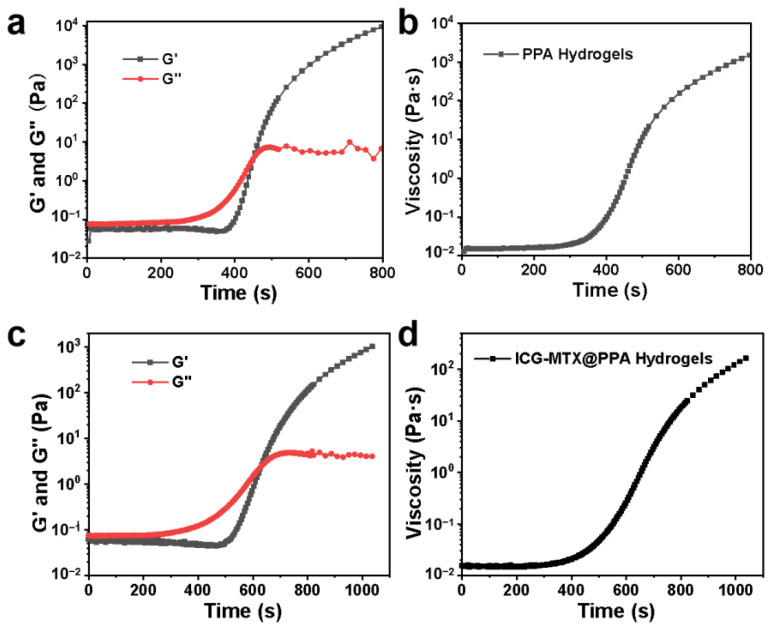
(**a**) The storage modulus (G′) and loss modulus (G″) of PPA Gels vary with time. (**b**) The viscosity of PPA Gels varies with time. (**c**) The storage modulus (G′) and loss modulus (G″) of ICG—MTX@PPA Gels vary with time. (**d**) The viscosity of ICG—MTX@PPA Gels varies with time.

**Figure 8 polymers-14-05547-f008:**
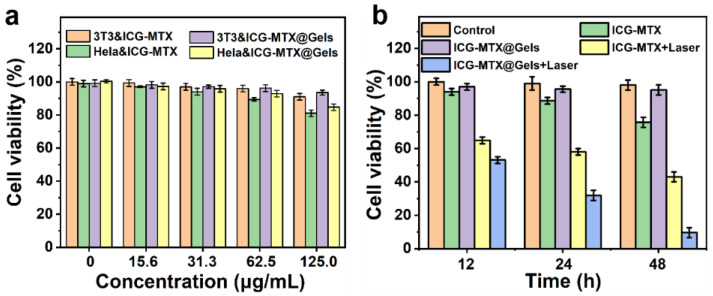
Analysis of in vitro cytotoxicity. (**a**) The cell viability of 3T3 and Hela treated with different concentrations of ICG—MTX and ICG—MTX@PPA Gels. (**b**) The cell viability of Hela cells treated with ICG—MTX with/without light and ICG—MTX@PPA Gels with/without light after 12, 24, and 48 h. The data are based on the average value ± S.D. Means (n = 5).

**Figure 9 polymers-14-05547-f009:**
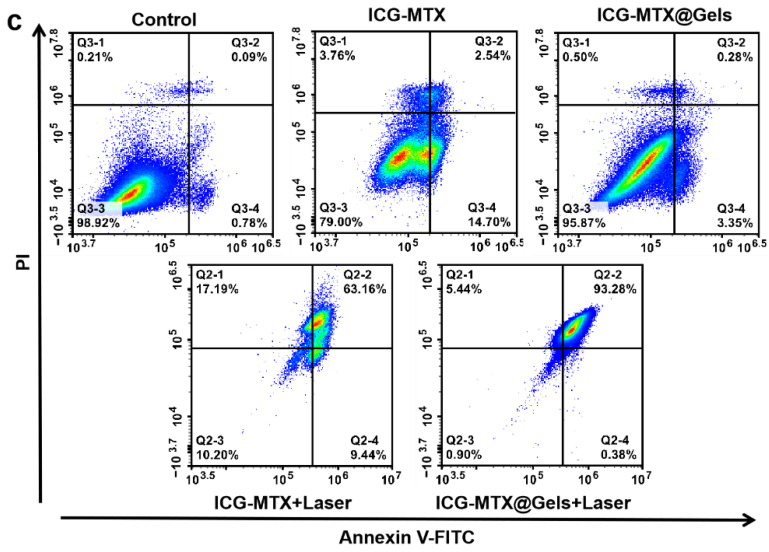
The survival of Hela cells cultured with ICG—MTX or ICG—MTX@PPA Gels for 24 h with or without laser irradiation (808 nm, 0.5 W·cm^−2^, 5 min) investigated by flow cytometry. I. PBS, II. ICG—MTX, III. ICG—MTX@PPA Gels, IV. ICG—MTX with laser, and V. ICG—MTX@PPA Gels with laser.

**Figure 10 polymers-14-05547-f010:**
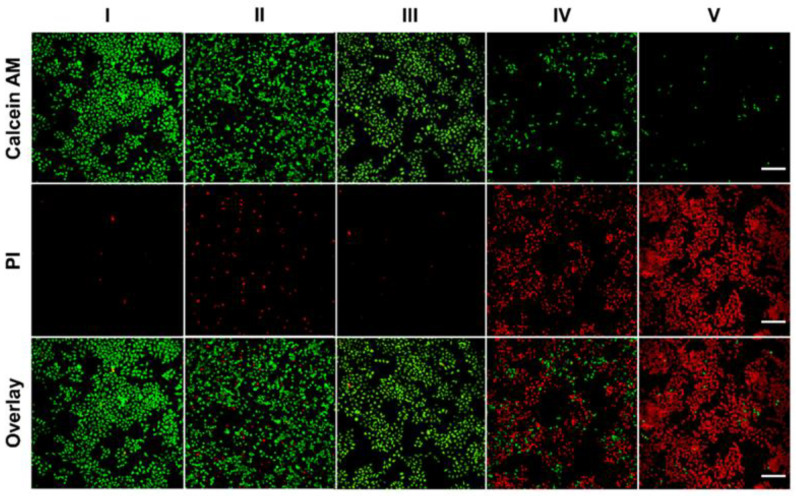
The fluorescence inverted microscope was used to take fluorescent images of Hela cells stained with Calcium-AM (living cells) and PI (dead/late apoptotic cells) under different conditions. Scale: 200 µm. I. PBS, II. ICG—MTX, III. ICG—MTX@PPA Gels, IV. ICG—MTX with laser and V. ICG—MTX@PPA Gels with laser.

## Data Availability

Not applicable.

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
