# Peer review of "Injectable Nanomedicine–Hydrogel for NIR Light Photothermal–Chemo Combination Therapy of Tumor"

_polymers, 2022, doi:10.3390/polym14245547_

Round 1

Reviewer 1 Report

Polymers 2065910

In this manuscript, the authors report on a nanoparticle-loaded hydrogel system for cancer treatment through combined photothermal and chemotherapies. The nanoparticle has been characterized and the photothermal performance of the latter was also discussed. Gel prepared and characterization was also presented. The nanoparticle was found to release at a slow rate from the hydrogels and in vitro experiments indicated that the irradiated hydrogels could effectively kills cancer cells.

Overall, the results presented in this paper are very interesting. There are only few queries to be addressed before it can be accepted for publication.

Major comments

1. Can the authors highlight and explain the ‘injectable’ aspect of the synthesized hydrogel.

2. Can the authors discuss more on the drug release kinetics?

3. Section 3.4: Can the authors measure the pore sizes of PPA and ICG-MTX@PPA hydrogels?

4. Figure 8: Please include % cell viability corresponding to the control after 24 and 48 hrs

5. From Fig 8a, it seems that the ICG-MTX has similar cytotoxicity to both 3T3 and HeLa cells. Have the authors investigated the % cell viability of 3T3 cells after treatment with light with hydrogels after 24 and 48 hrs?

6. It is recommended to investigate the cytotoxicity of the hydrogels on Hela cells when the latter is in co-culture with 3T3 cells so as to better mimic in vivo conditions.

7. In the Introduction, can the authors briefly discuss the specific type of cancer being targeted with this product and why?

Minor comments

1. Please define all abbreviations used in the text for example AIPH, PPA

2. Please check the language and grammar

Author Response

Major comments

  1. Can the authors highlight and explain the ‘injectable’ aspect of the synthesized hydrogel.

Response: Thank you for your valuable suggestion. The state of the hydrogel precursor solution is liquid, and it can be injected to tumor site through a syringe. The relevant content has been supplemented in lines 76 and 77, and corresponding references have been inserted in line 83.

  1. Can the authors discuss more on the drug release kinetics?

Response: This is the basic characteristic of hydrogels. The relevant content has been supplemented in lines 83 to 87, and references have been inserted.​ In addition, the contents of Figure 2d are supplemented.

  1. Section 3.4: Can the authors measure the pore sizes of PPA and ICG-MTX@PPA hydrogels?

Response: The pore sizes of PPA and ICG-MTX@PPA hydrogels have been supplemented in Figure 6. (Lines 281 and 290).

  1. Figure 8: Please include % cell viability corresponding to the control after 24 and 48 hrs.

Response: The corresponding percentage of cell viability of the control group after 24 and 48 hours has been supplemented in Figure 8b.

  1. From Fig 8a, it seems that the ICG-MTX has similar cytotoxicity to both 3T3 and HeLa cells. Have the authors investigated the % cell viability of 3T3 cells after treatment with light with hydrogels after 24 and 48 hrs?

Response: Figure 8a compares the dark toxicity comparison of hydrogel-treated 3T3 and Hela cells. In general, phototoxicity tests are not performed on normal cells.

  1. It is recommended to investigate the cytotoxicity of the hydrogels on Hela cells when the latter is in co-culture with 3T3 cells so as to better mimic in vivo conditions.

Response: With our existing technology, we can't cultivate 3T3 and Hela cells in the same petri dish at the same time. Even if the culture is appropriate, it is still difficult to distinguish the two types of cells during or after treatment.

  1. In the Introduction, can the authors briefly discuss the specific type of cancer being targeted with this product and why?

Response: Folate receptor is a cell surface glycoprotein highly expressed on cancer cells while its expression on normal cells is low or undetectable. Therefore, folate (FA) has been extensively investigated and applied for the selective delivery of therapeutics to cancers, including breast cancer, lung cancer, ovarian cancer, colorectal cancer, and head and neck cancer.

Since the structure of MTX is extremely similar to folic acid and can target folic acid receptors, this product is mainly aimed at tumor cells with folic acid receptors, ​such as Hela, 4T1 cells. The relevant content has been supplemented on line 86.

Minor comments

  1. Please define all abbreviations used in the text for example AIPH, PPA

Response: The information about AIPH has been supplemented on lines 78 to 81, and the description of the abbreviation of PPA is supplemented on lines 82 to 83.

  1. Please check the language and grammar

Response: The manuscript has been carefully checked and revised.

Reviewer 2 Report

In this paper, a targeted nanomedicine has been combined with injectable hydrogel for photothermal-chemotherapy combination therapy. The integration of ICG-MTX nanomedicines into hydrogel could introduce the NIR photothermal function of ICG-MTX. The nanomedicine could effectively remain at the site of the tumor and continue to be released from the hydrogel. The injectable nanomedicine-hydrogel system has a good therapeutic effect on tumors. The paper is in general well-organized and the conclusions are supported by the results. However, there are a few issues that should be addressed:

1. Which type of cancer could be targeted and inhibited by nanomedicine?

2. What is AIPH? Please give the full name of it, and show the function it plays.

3. Line 125, page 3, what does the lambda mean in equation (a), please explain it.

4. In Figure 1, the molecular formula is unclear, please revised.

5. In Figure 5b, the AIPH should be added to the Figure legends.

6. In Figure 8b, the viability of Hela cells of control groups after 24h and 48h are missing, please add them.

Author Response

  1. Which type of cancer could be targeted and inhibited by nanomedicine?

Response: Thank you for your valuable suggestion. Folate receptor is a cell surface glycoprotein highly expressed on cancer cells while its expression on normal cells is low or undetectable. Therefore, folate (FA) has been extensively investigated and applied for the selective delivery of therapeutics to cancers, including breast cancer, lung cancer, ovarian cancer, colorectal cancer, and head and neck cancer.

Since the structure of MTX is extremely similar to folic acid and can target folic acid receptors, this product is mainly aimed at tumor cells with folic acid receptors, such as Hela, 4T1 cells.​ The relevant content has been supplemented on line 86.

  1. What is AIPH? Please give the full name of it, and show the function it plays.

Response: The full name of AIPH and its functions have been supplemented in lines 78 to 81.

  1. Line 125, page 3, what does the lambda mean in equation (a), please explain it.

Response: The Aλ in formula (a) is explained on line 135, and the format of λ in formula has been revised.

  1. In Figure 1, the molecular formula is unclear, please revised.

Response: Figure 1 has been replaced with a high-definition image.

  1. In Figure 5b, the AIPH should be added to the Figure legends.

Response: The legend of Figure 5b has been modified.

  1. In Figure 8b, the viability of Hela cells of control groups after 24h and 48h are missing, please add them.

Response: The corresponding percentage of cell viability of the control group after 24 and 48 hours has been supplemented in Figure 8b.

Reviewer 3 Report

this study on NIR-responsive hydrogels is structured very well and the experimental design is thorough. the introduction clearly places this work within the field and the novelty is highlighted.

this work looks like a valuable innovation and should be of interest to many in the field. the data is presented quite nicely and the manuscript overall is a great example of high quality and impactful research conducted with appropriate depth and purposeful experimental design.

Author Response

Thank you for your positive comments.

Reviewer 4 Report

The article proposes a targeted nanomedicine formulation development combining a photothermal reactive with a chemotherapy drug incorporated into a hydrogel. Overall, the article is well designed and written, with only a few points to be considered:

- rheology experiments: please insert data from flow curves in order to discuss the effects of the conjugate ICG-MTX incorporation into hydogels stability 

- cumulative drug release experiments must be also performed considering the adequate temperature to fit it with photothermal performance.

- please provide information about release mechanisms mathematic modelling of release curves from ICG-MTX plain and ICG-MTX in hydrogels

- please present statistical analysis on Fig. 8b 

Author Response

  1. Rheology experiments: please insert data from flow curves in order to discuss the effects of the conjugate ICG-MTX incorporation into hydogels stability.

Response: Thank you for your valuable suggestion. The flow curve data has been shown in Figure 7 of the original text. We updated the discussion on the effect of the addition of conjugated ICG-MTX on the stability of the hydrogel from lines 313 to 318.

  1. Cumulative drug release experiments must be also performed considering the adequate temperature to fit it with photothermal performance.

Response: The drug release data at photothermal temperature has been supplemented in Figure 2d.

  1. Please provide information about release mechanisms mathematic modelling of release curves from ICG-MTX plain and ICG-MTX in hydrogels.

Response: PNIPAM hydrogel has no dissociation process in the treatment stage, and the release of the drug depends on permeation process. The relevant descriptions and references are on lines 85 to 88.

  1. Please present statistical analysis on Fig. 8b 

Response: The statistical analysis of Figure 8b has been supplemented from lines 346 to 353.

Round 2

Reviewer 1 Report

The manuscript can be accepted in the current form.